# Analysis of Global Sea Level Change Based on Multi-Source Data

**Yongjun Jia** [1,2,3], **Kailin Xiao** [4,*], **Mingsen Lin** [1] **and Xi Zhang** [1]

1  National Satellite Ocean Application Service, Beijing 100081, China
2  Southern Marine Science and Engineering Guangdong Laboratory (Guangzhou), Guangzhou 511458, China
3  Key Laboratory of Space Ocean Remote Sensing and Application (MNR), Beijing 100081, China
4  School of Marine Technology and Geomatics, Jiangsu Ocean University, Lianyungang 222005, China
*  Correspondence: 2019220268@jou.edu.cn

**Abstract:** Global sea level rise is both a major indicator and consequence of global warming. At present, global warming is causing sea level rise in two main ways: one is the thermal expansion of sea water, and the other is the injection of large amounts of fresh water into the ocean after glaciers and ice sheets melt. In this paper, satellite altimeter data are used to study the total changes of global sea level from 2002 to 2020. Different from most previous studies, this study proposes a calculation method of sea level anomaly using only the along track altimetry data, which is similar to considering the satellite points as tide gauges, in order to avoid the error caused by interpolation in the map data. In addition, GRACE satellite data are used to calculate the changes of global sea level caused by water increase; temperature and salinity data are used to calculate the changes from ocean thermal expansion. Next, using satellite altimetry data, the calculation results show that the global sea level rise rate in the period of 2002–2020 is 3.3 mm/a. During this period, the sea level change caused by the increase of sea water calculated with GRACE satellite data is 2.07 mm/a, and that caused by the thermal expansion of seawater is 0.62 mm/a. The sea level rise caused by the increase of water volume accounts for 62.7% of the total sea level rise.

**Keywords:** sea level change; satellite altimetry; ocean thermal expansion

## 1. Introduction

In recent years, with global warming, sea level rise has accelerated and has become a slow-onset marine disaster, which has gradually attracted more and more attention around the world [1–3]. In terms of eco-environmental protection, sea-level rise will seriously threaten coastal areas, especially the local ecosystem of sedimentary coasts and islands [2]. In addition, the issues of coastline erosion, island disappearance, and soil salinization will become more and more serious. In terms of social and economic development, sea level rise will cause irreparable economic losses to coastal agriculture and other economic industries, while some infrastructure in coastal areas and islands will be affected and damaged, disturbing the lives and property safety of local residents there.

There are many causes for the change of global sea level. The change of seawater temperature and salinity can lead to the change of seawater volume [4]. In addition, the melting of the Antarctic and Greenland ice sheets, the change of land water storage, and the evaporation of seawater will lead to geographically uneven sea level change. Added to this, the land subsidence caused by glacier equilibrium adjustment and plate movement will affect the change of regional and global average sea level. The two main factors affecting sea level change can be summarized as follows: mass sea level change caused by melt water from glaciers and ice sheets; and steric sea level change caused by thermal expansion of seawater as it warms [5].

Prior to the 1990s, the data used to calculate sea-level changes were mainly tide gauge data [6]. With the emergence of satellite altimetry, the use of TOPEX/Poseidon (hereinafter referred to as T/P) and Jason series satellites to study sea level change has gradually

increased. Through these means, we can obtain sea level change information with high resolution and high spatio-temporal information. In 2002, the National Aeronautics and Space Administration (NASA) of the United States and the German space flight center jointly launched the GRACE gravity satellite. The Earth's time-varying gravity field data can be used to calculate the mass change of global seawater [7]. Chambers et al. [8] estimated the global average sea level change by using Grace data for one and a half years from 2002 to 2003. The results are basically consistent with those calculated by the T/P data for the 11-year period of 1992–2003 and Jason-1 data for the period of 2002–2003, which verifies the availability of GRACE data in sea level research. Chen et al. [9] carried out a comprehensive review of GRACE/GRACE-FO satellite gravimetry, and discussed in detail several major challenges in using GRACE/GRACE-FO gravity measurements to study mass changes, and how we should address them. At the end of 1999, the United States, Japan, and other countries proposed a global ocean observation plan, which mainly uses Argo floats to monitor the temperature and salinity changes in the upper layer of seawater. At present, there are currently over 3900 floats in operation. Each float can provide more than 10 pieces of temperature and salinity profile data every year, greatly facilitating the calculation of the specific volume sea level change [10].

Many scientists have researched and analyzed the global sea level change and its inducing factors with the help of satellite altimetry data and satellite gravity data. Vinogradov et al. [11] analyzed the seasonal variation characteristics of global sea level for the period of 1992–2004 using satellite altimetry data, numerical model data, heat flux, fresh water flux, wind stress, and a series of data. Garcia et al. [12] studied and analyzed the annual variation characteristics of sea level in the Mediterranean using satellite altimetry data, Ecco ocean model data, and grace gravity satellite data. Su [13] used satellite altimetry data, GRACE gravity data, and temperature salinity data to analyze the contribution of specific volume sea level and seawater mass changes to global sea level changes. Feng et al. [14] used satellite altimetry data, gravity data, Argo temperature, salinity data, and tide gauge data to study the changes of global sea level from 2005 to 2013 on a seasonal and interannual scale. From altimetry, Argo, and GRACE/GRACE-FO data, Barnoud et al. [15] found that the global mean sea level budget is not closed after 2016, and the error in Argo salinity measurements provides ~40% improvement in the non-closure. Chen et al. [16] have found that the global ocean mass change obtained by GRACE/GRACE-FO is generally in good agreement with the estimates of Altimeter-Argo, while during the late-stage GRACE and GRACE-FO periods, the global mean ocean mass differences between GRACE/GRACE-FO and altimeter-Argo become larger and systematic. Many organizations or institutions have published global sea level rise products, such as NOAA, NASA, CSIRO, and Copernicus Marine Service [17–19]. These data are very important for the study of global sea level rise. When using satellite altimetry data to calculate sea level change, most of them use the map data after spatio-temporal interpolation. To make the work easier, the merged multi-satellite altimeter data sets are used to study global sea level change. In most map data used in global sea level changes calculation, the values of grid points are obtained by interpolation. The benefit of this is to ensure that sea level changes can be calculated at all points in the global sea area. In this study, the grid values are directly obtained by weighted average of the along track data, so as to avoid the error caused by interpolation.

This paper seeks to calculate the total sea level change with the help of satellite altimetry along track data, that is, all the along track sea level anomaly observations falling into the divided grid within the required time range are weighted averaged as the sea level anomaly of the grid, instead of introducing new errors through spatio-temporal interpolation. The idea of calculating global sea-level change from along track altimetry data is closer to that of tide stations. The satellite altimetry data also avoid the influence of topographic subsidence at the tide gauge station. China's altimetry satellite was not used in all studies. In this study, HY-2B and Jason series satellites with the same reference system were used to study global sea level rise. At the same time, the mass change of gravity satellite and the temperature and salinity data are used to calculate the sea level rise.

Section 1 mainly introduces the data used in this paper; Section 2 describes the method of calculating sea level change with different data; Section 3 presents the results of global sea level change and mutual inspection and analysis; and Section 4 is the discussion and conclusions.

## 2. Data

### 2.1. Satellite Altimetry Data

The altimeter data used in this study are mainly Jason-1/2/3 Geophysical Data Record (GDR) and HY-2B GDR data. Jason-1 contains 374 cycles before orbit change, while Jason-2 contains the first 18–327 cycles, and Jason-3 contains the first 33–177 cycles; finally, HY-2B contains the data of the first 2–71 repetition cycles.

At the same time, the T/P, Jason series, ERS/ENVISAT, HY-2 series, and other multi-source satellites are also used to integrate the global ocean grid SSALTO/DUACS sea surface height anomaly monthly average products, with a spatial resolution of $0.25 \times 0.25°$ and global coverage.

Table 1 shows the satellite altimetry data used in this research study.

**Table 1.** Satellite altimetry data.

| Satellite | Cycle Number | Time |
| --- | --- | --- |
| Jason-1 | 1–374 | January 2002 to December 2011 |
| Jason-2 | 18–327 | January 2009 to May 2017 |
| Jason-3 | 33–177 | January 2017 to November 2020 |
| HY-2B | 2–71 | December 2018 to September 2020 |
| Grid data | | January 2002 to February 2020 |

### 2.2. GRACE Satellite Data

The gravity satellite data used in this study are the TELLUS_GRAC-GRFO_MASCON_CRI_GRID_RL06_V2 global regional dataset of gravity anomalies (hereafter referred to as RL06M) published by the Jet Propulsion Laboratory (JPL) of the United States [20,21], which is based on the Earth mass values processed on the basis of GRACE and GRACE_FO satellite data at Level-1. The gridded global monthly mean water storage anomaly is processed by the mascon algorithm [22,23]. The mass change of seawater is equivalent to the change of sea surface height under the condition of temperature zero and salinity 35, which is presented in the form of equivalent water level. A coastal resolution improvement filter has been used to the data [24]. The influence of leakage error signals around the coastline thereby has effectively been removed. The spatial resolution of the data is $0.5 \times 0.5°$, and the time span is from April 2002 to December 2020, with a total of 192 months of data.

### 2.3. Temperature and Salinity Data

The temperature and salinity data used in this study are mainly the 3D ocean temperature and salinity dataset (MULTIOBS_GLO_PHY_TSUV_3D_MYNRT_015_012) published by the Copernicus Marine Environment Monitoring website (CMEMS), which is a reanalysis of monthly averaged data [25,26]. To date, there are about 4000 Argo floats operating across the globe. These floats are distributed over the global ocean to measure temperature and salinity. In this study, due to the inadequate global coverage of the early Argo floats, data from 2003 onward were used to calculate steric sea level change. The data selected span the period from January 2003 to December 2019, with a spatial resolution of $0.25 \times 0.25°$, and are divided into a total of 50 vertical standard layers which can reach 5500 m in depth, as shown in Table 2 below.

**Table 2.** Temperature and salinity data.

| Level Number | Depth (m) | Level Number | Depth (m) | Level Number | Depth (m) | Level Number | Depth (m) | Level Number | Depth (m) |
|---|---|---|---|---|---|---|---|---|---|
| 1 | 0 | 11 | 50 | 21 | 175 | 31 | 550 | 41 | 1500 |
| 2 | 5 | 12 | 55 | 22 | 200 | 32 | 600 | 42 | 1750 |
| 3 | 10 | 13 | 60 | 23 | 225 | 33 | 700 | 43 | 2000 |
| 4 | 15 | 14 | 65 | 24 | 250 | 34 | 800 | 44 | 2500 |
| 5 | 20 | 15 | 70 | 25 | 275 | 35 | 900 | 45 | 3000 |
| 6 | 25 | 16 | 80 | 26 | 300 | 36 | 1000 | 46 | 3500 |
| 7 | 30 | 17 | 90 | 27 | 350 | 37 | 1100 | 47 | 4000 |
| 8 | 35 | 18 | 100 | 28 | 400 | 38 | 1200 | 48 | 4500 |
| 9 | 40 | 19 | 125 | 29 | 450 | 39 | 1300 | 49 | 5000 |
| 10 | 45 | 20 | 150 | 30 | 500 | 40 | 1400 | 50 | 5500 |

## 3. Sea Level Change

Global sea level changes can be attributed to the sea water mass changes and steric sea level change [27,28]. The steric sea level change is mainly a result of the expansion and contraction of seawater volume caused by the temperature and salt changes of seawater, which can be estimated from the measurement data. Meanwhile, the mass sea level change is mainly caused by the melting of land glaciers and ice sheets, which can be calculated from the GRACE gravity satellite data. The relationship can usually be expressed by the following [10,29]:

$$SLC_{total}(\theta, \lambda, t) = SLC_{barystatic}(\theta, \lambda, t) + SLC_{steric}(\theta, \lambda, t) \tag{1}$$

where $SLC_{total}$ represents the overall sea level change; $SLC_{barystatic}$ is the sea level change due to changes in seawater mass; $SLC_{steric}$ is the steric sea level change; $\theta$ denotes the longitude range of the region under study; $\lambda$ denotes the latitude range of the region being studied; and $t$ denotes the size of the time span over which the sea level changes are studied.

### 3.1. Satellite Altimeter Data Processing

This study is concerned with dynamic sea level change, rather than absolute sea level height. The sea level anomalies are shown in Equations (2) and (3):

$$Sea\ Surface\ Height = Satellite\ Altitude - Altimeter\ Range - Corrections \tag{2}$$

$$Sea\ Level\ Anomaly = Sea\ Surface\ Height - Mean\ Sea\ Surface \tag{3}$$

As the measurement is affected by the troposphere, ionosphere, atmosphere, sea state, tides, etc., the measurement data must be corrected accordingly [30]. Usually, the reference ellipsoids among missions may be different, and models or algorithms used to deal with various error corrections are also not the same. Therefore, in order to effectively eliminate the calculation differences, this paper unifies the calculation basis and processing methods of each error correction among satellite missions. For example, the reference ellipsoid of all satellite missions used is unified as a WGS84 ellipsoid, and the dry troposphere correction is calculated based on the same data provided by European Center for medium range weather forecasting (ECMWF), etc. After unifying the reference ellipsoid and the processing benchmark of each error correction among satellite missions, the correction can be calculated as shown in Equation (4) [31]:

$$h_{Corrections} = h_{iono} + h_{dry_{trop}} + h_{wet_{trop}} + h_{ssb} + h_{inv_{bar}} + h_{otd} + h_{ptd} + h_{std} + h_{hf_{fluc}} \tag{4}$$

where $h_{iono}$ is the ionospheric correction; $h_{dry_{trop}}$ is the dry troposphere correction; $h_{wet_{trop}}$ is the wet troposphere correction; $h_{ssb}$ is the sea state deviation correction; $h_{inv_{bar}}$ is the atmospheric inverse pressure correction; $h_{otd}$ is the ocean tide; $h_{ptd}$ is the polar tide; $h_{std}$ is the solid tide; and $h_{hf_{fluc}}$ is the high frequency oscillation. In addition, the mean sea level is the average of sea level heights relative to the ellipsoid over many years and is calculated

on a regular grid and processed by combining data from all of the tide gauge stations and satellite altimetry. Reference corrections are shown in Table 3.

**Table 3.** Reference corrections overview.

|  | Jason-1 | Jason-2 | Jason-3 | HY-2B |
|---|---|---|---|---|
| Orbit | POE-E | POE-F | | POE-D |
| Ionospheric | Filtered dual-frequency altimeter range measurements | | | |
| Sea state bias | Non parametric | | | |
| Wet troposphere | JMR radiometer | AMR radiometer | | CMR radiometer |
| Dry troposphere | ERA5 model based | | | |
| Atmospheric inverse pressure correction | Inverse barometer low frequencies | | | |
| High frequency oscillation | TUGO forced with analyzed ERA5 | | | |
| Ocean tide | FES 2014 | | | |
| Pole tide | Desai et al., 2015 [32] | | | |
| Solid tide | Cartwright and Tayler, 1971 [33] | | | |
| Mean sea surface | CNES/CLS 2015 | | | |

In this paper, due to the respective limitations of satellite technology, geography, and environment, only the changes within 66° north-south latitude and 180° east-west longitude are discussed in the calculation of sea level changes using satellite radar altimeter observations. While calculating the sea surface anomaly, the along track altimetry data are directly used without interpolation. The basic idea is to first divide the Earth's surface between 66° north and south latitude and 180° east and west longitude into a $1 \times 1°$ grid, as shown in Figure 1. When the ground tracks of satellites are in a $1° \times 1°$ grid, calculate the average value of sea surface height anomaly of satellite observations in this grid. Figure 2 shows the ground tracks of satellites of Jason-1. Finally, the monthly, quarterly, or annual mean product of sea level anomalies for the study area is obtained by averaging the values over the desired time range. Since the regular lat/lon grids are used, the grid spacing (or area) changes with latitude. Thus, a weighted average to derive the global mean is necessary. The regional weighted averaging method used in this study is as follows:

$$\overline{h_t} = \frac{\sum_\lambda \sum_\phi h_{\lambda\phi t} \cos(\phi)}{\sum_\lambda \sum_\phi \cos(\phi)} \tag{5}$$

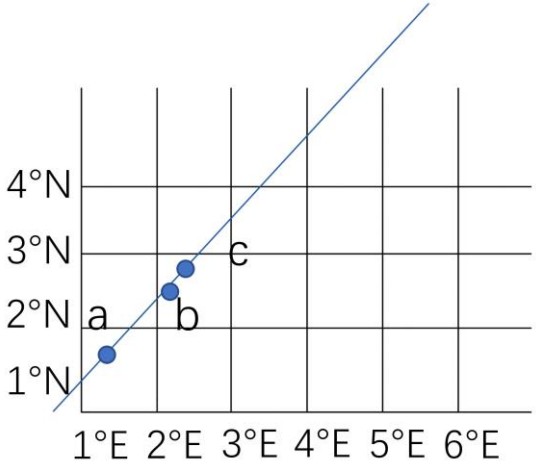

**Figure 1.** Schematic diagram of the grid.

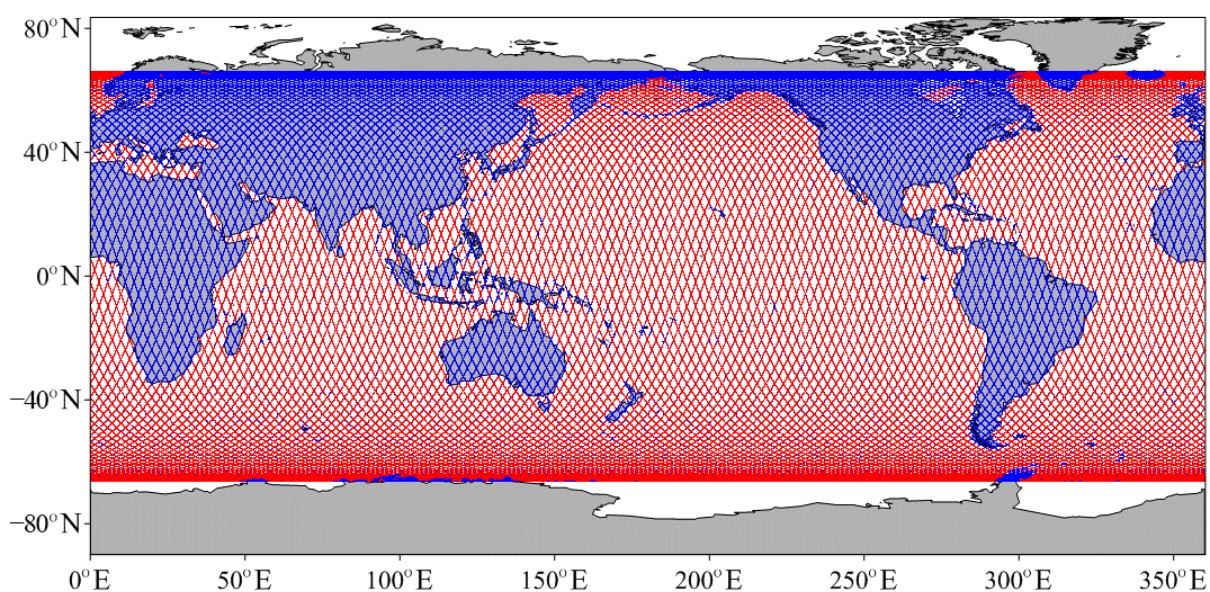

**Figure 2.** Ground tracks of the Jason−1 satellite.

The variables are as follows: $\lambda$ is longitude, $\phi$ is latitude, $t$ is time, $\overline{h_t}$ is the regional average at time $t$, and $h_{\lambda\phi t}$ is the data value at these latitude and longitude at time $t$.

Like NOAA, CMEMS, CSIRO, and other organizations' methods of calculating global sea level rise, we also use map data to calculate it. The monthly average grid product for the map data is applied to calculate the anomalous sea level changes in the study area. In order to obtain a more objective and accurate picture of sea level rise, the calculation process is as follows:

(1) The $0.25 \times 0.25°$ monthly mean sea level anomaly was extracted from the ocean land mask file in the corresponding ocean study area for the period of January 2002 to February 2020.

(2) The sea level anomalies for each month were obtained by regional averaging over the marine study area, after which a time series was generated.

### 3.2. Analysis of Satellite Altimetry Results

GDR data from the HY-2B and Jason series were processed using the above method, with sea level products from SSALTO/DUACS. The monthly averaged plots of global sea level anomaly changes were then calculated, and are shown in Figure 3. The sea level changes are analyzed in the paper using time series analysis methods.

In the calculation of sea level rise rates, the most commonly used methods are mainly the linear model $y = ax + b$ and the stochastic dynamic model $y = A_1 \cos(w_1 t - \varnothing_1) + A_2 \cos(w_2 t - \varnothing_2) + Bt + C + \varepsilon_1(t) + \varepsilon_2(t)$. The linear model mainly uses the least squares method to solve for the slope of the rising trend line. In order to retaining the original sea level rise information calculated, this paper uses a linear regression least squares fitting algorithm to calculate the sea level change.

Next, the rise rate of global sea level change from 2002 to 2020 was obtained as 3.3 mm/a, as shown in Figure 4. This result is slightly different from previous research results [34–37]. At the same time, it is also slightly lower than the global sea level rise rate [38] announced by many international institutions, such as CU, AVISO, CSIRO, and NASA GSFC. One explanation for this is that the linear model used in the paper ignores the effect of interdecadal variability.

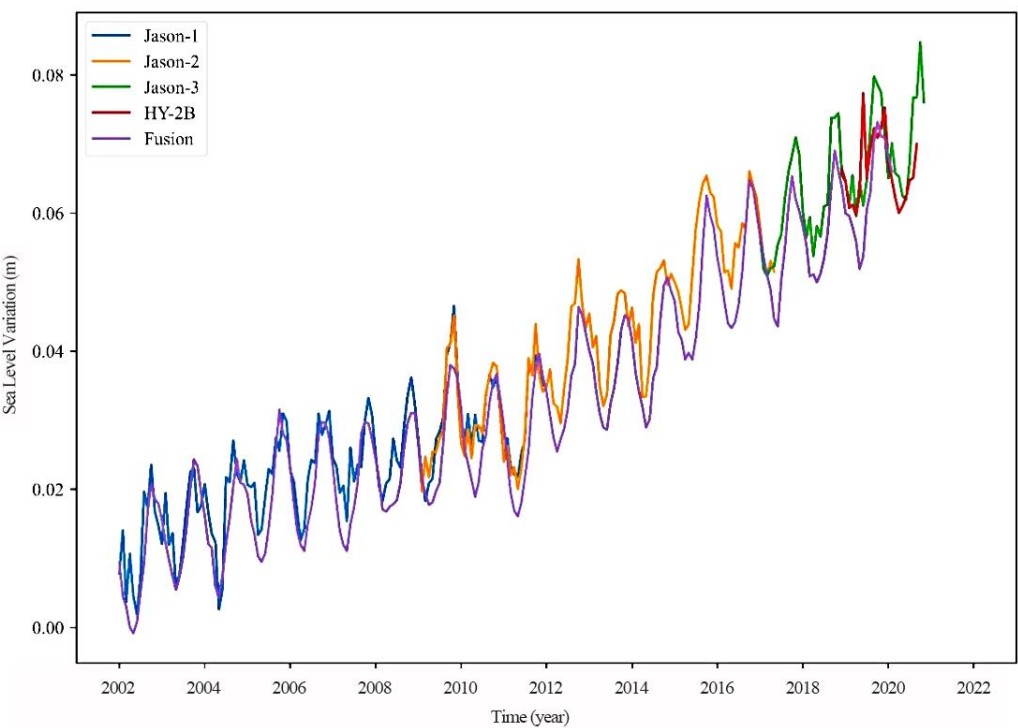

**Figure 3.** Time series of global sea level change anomalies, 2002–2020.

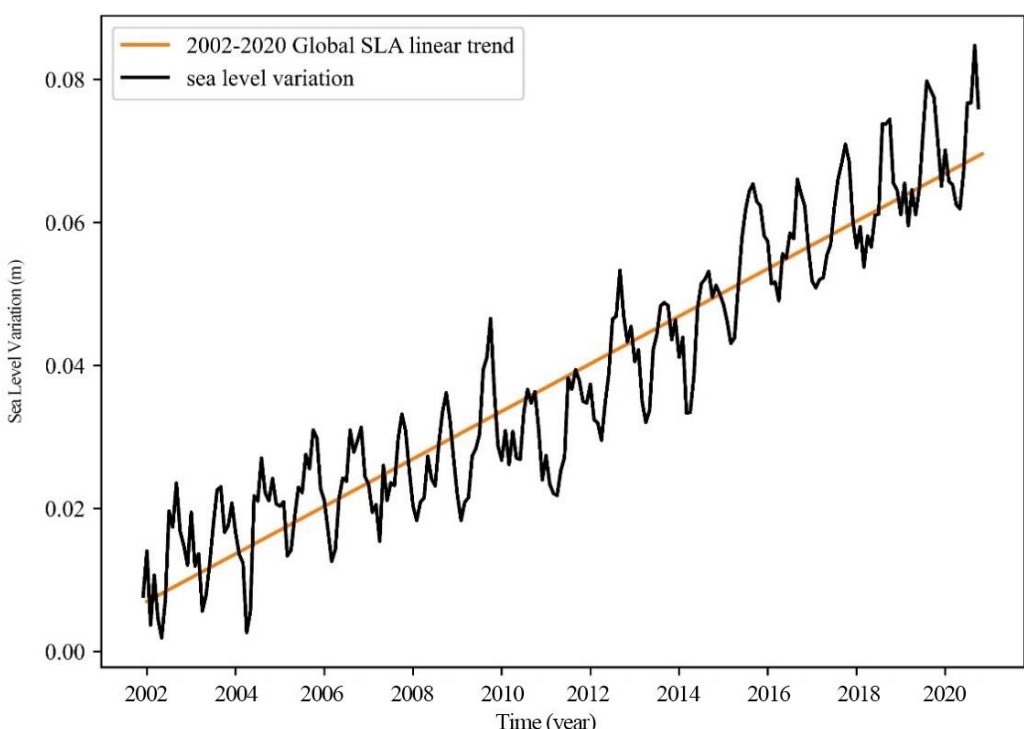

**Figure 4.** Analysis of global sea level change trends 2002–2020.

Next, the univariate quadratic regression equation is used to solve the acceleration of sea level rise. The acceleration of global mean sea level rise between 2002 and 2020 is shown to be 0.09 mm/a. This is much higher than the acceleration of global sea level rise in the nineteenth century of 0.0092 mm/a calculated by Chen [39], and also somewhat higher than the acceleration of global sea level rise in the last 25 years of 0.084 mm/a calculated by Nerem et al. [36]. This shows that the rate of sea level rise has been accelerating in the

past ten years. In order to more clearly illustrate the rate of sea level rise, this paper also divides 2002 to 2020 into two separate decadal periods, 2002–2012 and 2010–2020, and fits the global sea level rise rate in the first decade of 2002–2012 to be 2.4 mm/a, which is slightly lower than the overall rate of rise. The global sea level rise rate in 2010–2020 is 4.1 mm/a. The rate of sea level rise for 2010–2020 is 4.1 mm/a, which is much higher than the average rate of rise from 2002 to 2020, and this is another indication of the increasing rate of sea level change over the past decade.

From Figure 4, it can be observed that the global sea level anomaly has a clear annual cycle of variability, with oscillations of around 7 mm, and there is a clear seasonal signal variation. For the seasonal variation, its four seasons are considered: spring (March to May), summer (June to August), autumn (September to November), and winter (December to February). The rate of sea level rise is calculated for each season using a linear model, as shown in Table 4.

**Table 4.** Global sea level rise rate in the four seasons (mm/a).

| Season | Spring | Summer | Autumn | Winter |
|---|---|---|---|---|
| Rate of rise | 3.28 | 3.29 | 3.4 | 3.15 |

Due to the restrictions of the data, the winter calculation is only performed up to 2019. As seen in the table, the global sea level change is very close to the rate of rise in all four seasons, at 3.3 mm/a. It can also be seen from the trend graph that, when the seasonal signal is eliminated from the global sea level time series, the two are very close, thus indicating that the seasonal signal is a significant factor in the periodicity of sea level change.

### 3.3. Gravity Satellite Data Processing

The satellite altimetry data are restricted to between 66° north and south latitude, due to the geographical distribution of the observations. For the sake of consistency between the two, seawater mass variations between 66° north and south latitudes were also calculated using the RL06M data. In addition, the GRACE gravity field model has relatively low accuracy in the lower order terms, where the C20 (degree 2 order 0) term can have a non-negligible effect on the inversion results, and the satellite laser ranging (SLR) observations have a much lower satellite orbit. Meanwhile, the SLR data are more sensitive to the C20 term and have a higher accuracy compared to the C20 term in the RLO6M data. On the other hand, the C20 from the SLR solution was used to replace the C20 from the GRACE solution [40]. The first-order term was replaced using the value calculated by Swenson et al. [41]. In addition, the glacial equilibrium adjustment was measured, which affects the calculated gravity field results and thus must be deducted using the ICE6G-D model. Real-world geophysical information was also introduced as a means of removing the associated errors, by applying an improved filter CRI (Coastline Resolution Improvement) to reduce the effects of leakage errors caused by signal leakage from the ocean and land close to the periphery.

In the present paper, the monthly averaged products of the RL06M dataset are applied to extract equivalent water height data for the corresponding marine study area. Note that there is a gap between GRACE and GRACE –FO of 1 year. In order to ensure the continuity of the time series, the results of the missing months were interpolated using cubic spline interpolation, in order to obtain the equivalent water height for the complete time period. This was then converted into a corresponding time series plot of sea level change due to seawater mass.

### 3.4. Changes in Seawater Mass

The RL06M data were processed using the above method to obtain a monthly averaged time series plot of seawater mass change, as shown in Figure 5. The same linear regression least squares method was then selected to calculate the long-term trend of sea level change

due to seawater mass from 2002 to 2020, which is 2.07 mm/a, as shown in Figure 5. Next, the rate of increase was calculated for the same period of ten years before and after the sea level change trend due to seawater mass from 2002 to 2012 is 1.9 mm/a, accounting for 79.1% of the overall sea level rise rate in the same time period. This result is similar to the sea level change trend due to seawater mass from January 2004 to December 2010 calculated by Li [42], which is 1.84 mm/a, accounting for 73% of the total sea level rise rate. The sea level change trend due to seawater mass from 2010 to 2020 is 2.3 mm/a, accounting for 56.09% of the overall sea level rise rate in the same time period, which is a relatively small percentage. The main consideration is that, from 2011 onward, there are numerous missing months, and part of the data was obtained by interpolation, thus the data may deviate from the actual situation. Overall, however, sea level change due to seawater mass from 2002 to 2020 accounts for 67.8% of the overall upward trend.

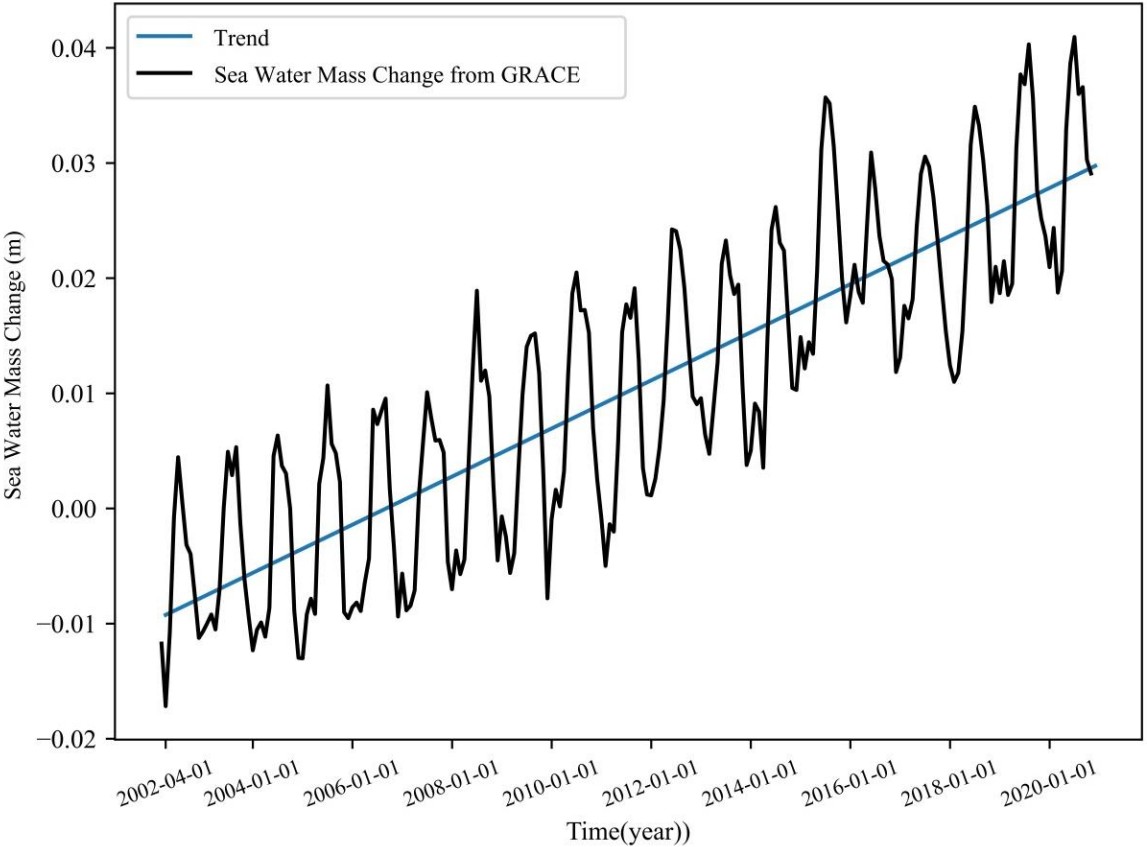

**Figure 5.** Linear trend in sea level change due to changes in sea water mass.

It can also be seen from the graph that the variation in seawater mass from the GRACE gravity satellite clearly shows the seasonal variation in the water cycle among the atmosphere, land, and ocean. It can also be seen that the minimum value of seawater mass for each year is generally at the beginning of January and February, then in August, when the temperature rises in the summer, causing snow, melting glaciers, and increased rainfall. This is the time when seawater is most abundantly stored, and its mass is greatest.

### 3.5. Steric Sea Level Calculation

The variations of ocean temperature and salinity lead to the sea level change, which is known as steric sea-level change. Steric change is one of the important factors of global sea

level change, and reflects the response of the ocean to global warming [43–45]. The steric sea-level change is calculated as follows:

$$\Delta\phi(\varphi,\lambda) = \int_0^h \frac{\rho_0(\varphi,\lambda,z) - \rho(\varphi,\lambda,z,t)}{\rho_0(\varphi,\lambda,z)} dz \qquad (6)$$

where $\varphi$ is the latitude, $\lambda$ is the longitude, z is the depth, $\rho$ is the real-time density at $(\varphi,\lambda)$, $\rho_0$ is the global standard seawater density, and $h$ is the maximum depth of integration.

As the variations of seawater temperature and salinity are very small in deep sea with a water depth of more than 1 km, and the measurement depth limit of Argo and other measuring instruments is also limited; in this study, the H value is taken as 1 km. For each layer, the calculated steric change is superimposed to obtain the overall steric sea level change value. The choice of H will affect the derived results. Especially when studying regional sea rather than global sea, this impact may be greater [46].

One of the calculations of seawater density involved in Equation (6) is based on the International State Equation of Seawater, which exhibits the relationship between the density of seawater and the salinity of sea water $S$, the temperature $T$, and applied pressure $P$, as shown in Equation (7) [47]:

$$\rho(S,T,P) = \rho(S,T,0)/[1 - P/K(S,T,P)] \qquad (7)$$

where $P$ is the applied or gauge pressure, $K$ is the Secant bulk modulus, and $\rho(S,T,0)$ is the standard seawater density according to International one-atmosphere Equation of State (1980).

### 3.6. Steric Sea Level Change

The study processes temperature and salinity data according to the method presented in Section 3.5, and calculates the change in global steric sea level anomalies. This is then extrapolated to 2020 in order to match the satellite altimetry and GRACE data, as shown in Figure 6.

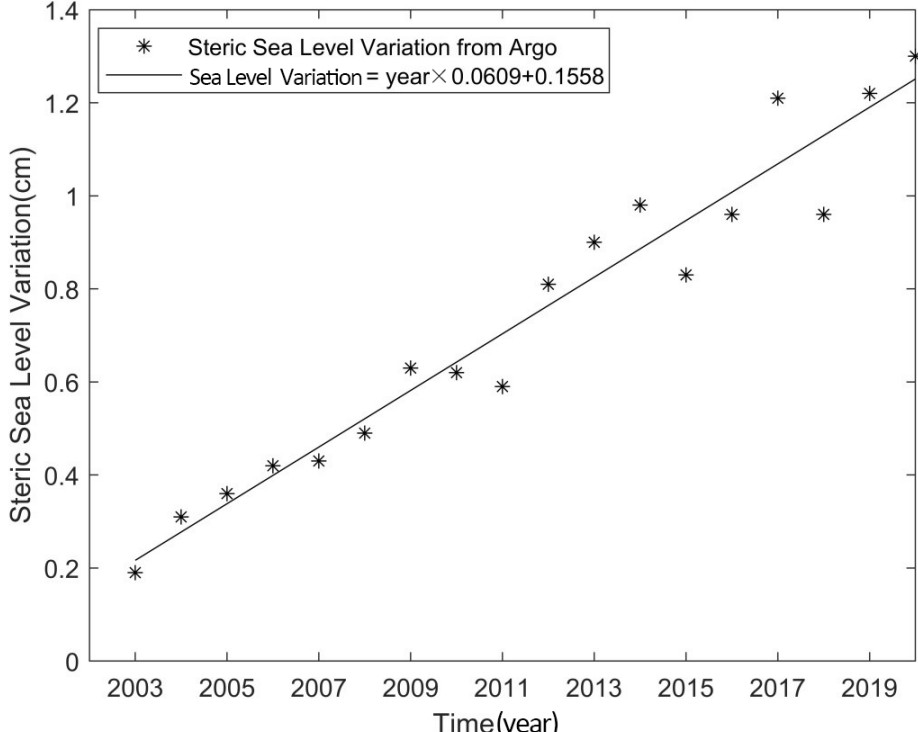

**Figure 6.** Changes in global steric sea level anomalies.

The linear model is fitted to obtain a long-term trend of 0.6 mm/a in sea level change due to seawater temperature and salinity change from 2003 to 2020, as shown in Figure 6. Once again, the rate of rise was calculated for the preceding and following decadal time periods, yielding a steric sea level trend of 0.58 mm/a for the period of 2003–2012, accounting for 24.16% of the overall rate of sea level rise over the same time period; and a steric sea level trend of 0.61 mm/a for 2010–2020, accounting for 14.8% of the overall rate of sea level rise over the same time period, which is a relatively small percentage. The main reason for this is the bias in the Argo salinity data from 2016 onward, with some of the interpolated results possibly deviating from the actual results. Overall, the change in steric sea level from 2002 to 2020 accounts for 19.6% of the overall rising.

## 4. Comparison of Results and Correlation Analysis

### 4.1. Comparison of the Two Results

According to Equation (1), the time series of global total sea level change from 2003 to 2020 are calculated. The global seawater mass changes from 2003 to 2020 are calculated using RLO6M. Using the global ocean temperature and salt reanalysis data, the time series of steric sea level changes in the global ocean are obtained. The contribution of seawater mass change and steric sea level change to global sea level rise is compared with global total sea level rise obtained by satellite altimetry. The results are shown in Figure 7. It is evident that the two-time series match relatively well on a global scale, with both showing very clear seasonal cyclical variations.

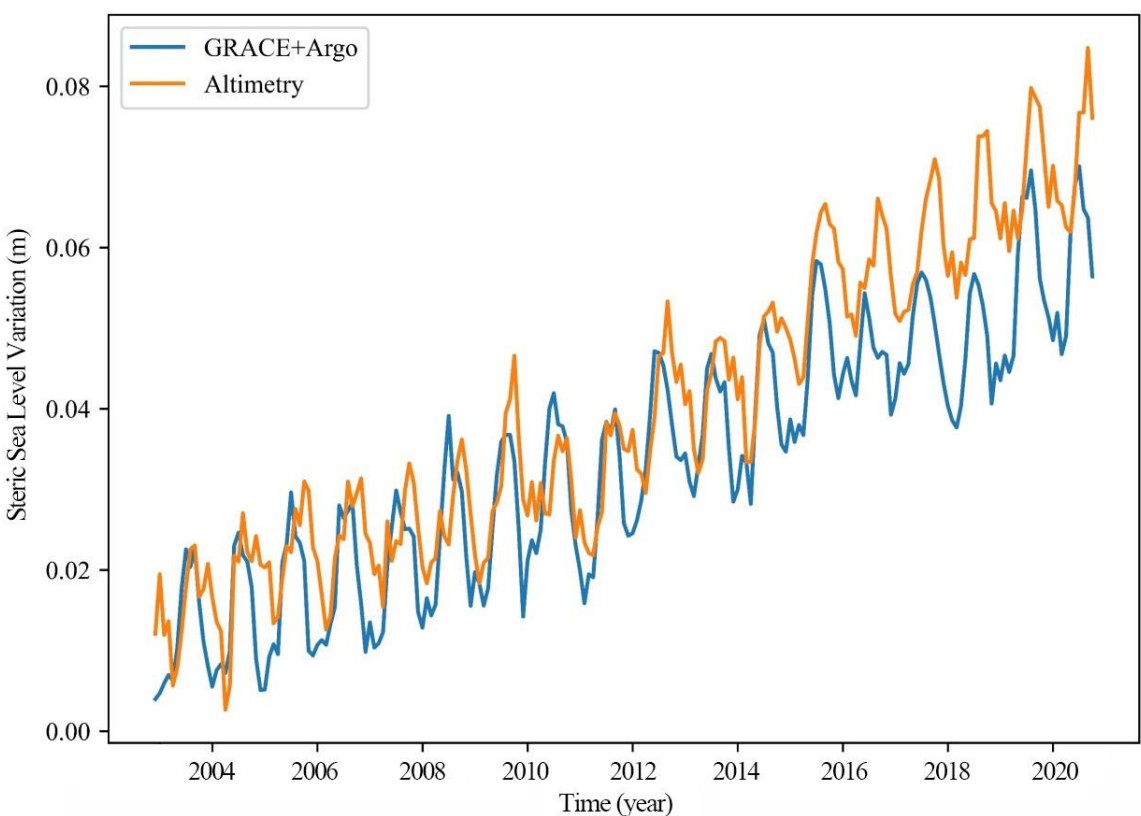

**Figure 7.** Global sea level change from 2003 to 2020, satellite altimetry, GRACE, and Argo.

It can be seen that the two calculated global sea level changes are in higher agreement and can verify this when analyzing sea level changes on a large global scale. It is also shown that the seawater mass change and the steric sea level change contribute to most of the sea-level rise. However, there are also some discrepancies, particularly in the time series after 2015, Figure 7. The main factors of this are as follows: (1) the errors carried by

the two measurements themselves; (2) the GRACE gravity satellite data are not complete and some of the data are missing; and (3) the temperature and salinity data used in the paper are generated by reprocessing the ocean float data, and there is still a large part of the ocean that is not monitored, due to the uneven global coverage of float locations [48].

### 4.2. Relevance Analysis

The mathematical formula proposed by renowned British statistician Carl Pearson is used to analyze the correlation between various data, which is one of the simpler methods of coefficient analysis in statistics. It is typically denoted by r and takes values between [–1, 1]. The correlation coefficient between two sets of data can be defined as follows [49]:

$$r = \frac{\sum_{i=1}^{n}(x_i - \overline{x})(y_i - \overline{y})}{\sqrt{\sum_{i=1}^{n}(x_i - \overline{x})^2 \sum_{i=1}^{n}(y_i - \overline{y})^2}} \tag{8}$$

The variables to be studied are $x$ and $y$; $x_i$ and $y_i$ in the formula are observed values, where I = 1, 2, 3,......, $n$. $\overline{x}$ and $\overline{y}$ are mean values calculated from observations. $x$ and $y$ are not directly related to the degree of variation, change in magnitude, or metric relationship presented between the relevant properties of the variables. In addition, as an expression of the properties between the two variables, $r$ is crucial. $r$ indicates the overall correlation between the two sets of data: when $r$ is positive, then the two sets of data are positively correlated; when $r$ is negative, the two sets of data are negatively correlated; and when $r$ is zero, the two sets of data are not directly related. According to Pearson's correlation formula, the correlation between several groups of data involved in the text is calculated.

The correlation between on-orbit and fused data, that between sea level change from the satellite altimeter and sea water mass change, and that between sea level change from satellite altimetry and GRACE+Argo, are respectively calculated according to Equation (8).

The correlation coefficient between the along-track and fused data is 0.972, with an extremely strong correlation between the two, as shown in Table 5.

**Table 5.** Correlation analysis (along-track data and fused data).

|  |  | Along-Track Data | Grid Data |
|---|---|---|---|
| Along-track data | Pearson Correlation | 1 | 0.972 |
|  | Significance (two-tailed) |  | 0.0 |
|  | Number | 227 | 227 |
| Grid data | Pearson Correlation | 0.972 | 1 |
|  | Significance (two-tailed) | 0.0 |  |
|  | Number | 227 | 227 |

The Pearson correlation coefficient between satellite altimetry and GRACE is 0.8, which is a good correlation. This is shown in Table 6.

**Table 6.** Correlation analysis (satellite altimetry and GRACE).

|  |  | Satellite Altimetry | GRACE |
|---|---|---|---|
| Satellite altimetry | Pearson Correlation | 1 | 0.8 |
|  | Significance (two-tailed) |  | 0.0 |
|  | Number | 224 | 224 |
| GRACE | Pearson Correlation | 0.8 | 1 |
|  | Significance (two-tailed) | 0.0 |  |
|  | Number | 224 | 224 |

The Pearson correlation coefficient among satellite altimetry, gravity and temperature and salt is 0.915, which is a good correlation. This is shown in Table 7.

**Table 7.** Correlation analysis (satellite altimetry, gravity and thermohaline).

|  |  | Satellite Altimetry | GRACE and Argo |
|---|---|---|---|
| Satellite altimetry | Pearson Correlation | 1 | 0.915 |
|  | Significance (two-tailed) |  | 0.0 |
|  | Number | 215 | 215 |
| GRACE and Argo | Pearson Correlation | 0.915 | 1 |
|  | Significance (two-tailed) | 0.0 |  |
|  | Number | 215 | 215 |

## 5. Conclusions

In this paper, we used satellite altimeter data, gravity satellite GRACE data and temperature and salinity data to study global sea level change between 66° degrees north and south latitudes from 2002 to 2020, so as to obtain the overall sea level change, seawater mass change, and steric sea level change, respectively. The global sea level change budget was analyzed. The three measurements are relatively well matched in the global-scale study, and also coincide well on seasonal scales. The barystatic sea level and the steric sea level are not constant in terms of its secular trend. From 2002 to 2020, the sea level rise caused by seawater mass change was 2.07 mm/a, accounting for 62.7% of the average sea level rise rate (3.3 mm/a), and the steric sea level change was 0.6 mm/a, accounting for 18.6% of the average sea level rise rate. Overall, seawater mass change is shown to be the main cause of sea level rise over the past 20 years. With the increasing of quality and spatial resolution of the GRACE satellite data, the use thereof to observe global seawater mass changes holds great promise. It is anticipated that continuous observations of global sea level and its mass component can be achieved on longer time scales, which is important for studying the links between these factors and global sea level change. These factors include glacial melting on snow-capped mountains, changes in the mass of polar ice caps, changes in water storage on land, and the relationship between sea level change and climate change.

The calculation of the global mean sea level (GMSL) was based on multi-mission satellite altimetry. There was similar research proposed with somewhat different results in recent years using the similar data and different data processing methods [50–53]. In these studies, GRACE data and Argo floats data were used to measure the water mass exchange and the steric component of global mean sea level, respectively. Overall, the global average sea level rise rate of this study is slightly lower than the results of similar studies but closer to the average of these studies [36,37,54]. As for the proportion of the water mass exchange and the steric component in the global sea-level budget, research results are different.

The comparison between the along track altimetry data and the fusion data shows that, when calculating the global sea level change, directly using the fusion altimetry data will cause the calculated sea level rise speed to be less than the real rise speed. It can also partly explain why the sea level rise caused by seawater mass change and steric sea level change is smaller than the actual sea level rise.

There remain major technical shortcomings in the use of GRACE satellites for seawater mass changes, and there is no ideal solution for the problem of signal leakage from the land to the sea to improve the signal in the ocean. In the follow-up satellite GRACE-FO, while it carries a laser ranging device, as of yet it has not been put into use, and the data currently measured are still microwave ranging, which will reduce the accuracy of the satellite. Second, the use of floats in the deep sea with depths greater than 2000 m is unable to obtain temperature and salinity data. Therefore, it is not possible to accurately obtain the desired amount of deep sea temperature and salinity changes on sea level changes through temperature and salinity data. Third, and finally, there remains uncertainty regarding the proportion of human activities in global warming, while how to effectively reduce human intervention and carbon emissions is also a key factor.

**Author Contributions:** Conceptualization, Y.J. and M.L.; methodology, Y.J.; validation, K.X. and Y.J.; investigation, M.L.; resources, Y.J.; data curation, X.Z.; writing—original draft preparation, Y.J. and K.X.; writing—review and editing, Y.J and M.L.; funding acquisition, M.L. All authors have read and agreed to the published version of the manuscript.

**Funding:** This research was funded by the National Natural Science Foundation (42192561, 42192531) and the Key Special Project for Introduced Talents Team of Southern Marine Science and Engineering Guangdong Laboratory (Guangzhou) (GML2019ZD0302).

**Data Availability Statement:** Not applicable.

**Acknowledgments:** The Jason-1/2/3 GDR-D and grid data were obtained from AVISO. The HY-2B data were obtained from https://osdds.nsoas.org.cn/#/ (accessed on 18 December 2021). The temperature and salinity data were obtained from https://resources.marine.copernicus.eu/products (accessed on 20 September 2020). The GRACE satellite data were downloaded from https://grace.jpl.nasa.gov/data/get-data/ (accessed on 27 August 2020).

**Conflicts of Interest:** The authors declare no conflict of interest.

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
