# Peer review of "Analysis of Global Sea Level Change Based on Multi-Source Data"

_remotesensing, doi:10.3390/rs14194854_

Round 1

Reviewer 1 Report (Previous Reviewer 1)

This paper has been thoroughly revised based on the suggestions. I think it is suitable to be published at Remote Sensing with minor changes:

Line 19, change ‘and temperature and salinity data are used to ocean thermal expansion’ to ‘ temperature and salinity data are used to calculate the changes from ocean thermal expansion’.

Line 124. ’ …monthly mean water reserve anomaly’, here ‘reserve’ should be 'storage'?

Author Response

This paper has been thoroughly revised based on the suggestions. I think it is suitable to be published at Remote Sensing with minor changes:

  • Line 19, change ‘and temperature and salinity data are used to ocean thermal expansion’ to ‘temperature and salinity data are used to calculate the changes from ocean thermal expansion’.

Response:We are grateful for the suggestion. To be more clearly and in accordance with your concerns, we have replaced “and temperature and salinity data are used to ocean thermal expansion” with “temperature and salinity data are used to calculate the changes from ocean thermal expansion”.

  • Line 124. ’ …monthly mean water reserve anomaly’, here ‘reserve’ should be 'storage'?

Response:Thank you for your careful work. We have modified this expression.

Reviewer 2 Report (Previous Reviewer 2)

The revised manuscript contains important information and is better organized than in the first version. There are some linguistic improvements in the revised version.

In my opinion, the work is ready for printing after taking into account minor imperfections that I notice:

Line 202: In the figure caption, please note that there is no capital letter and unnecessary space.

Figure 7: Poor drawing quality. Although the axis descriptions should be improved.

Line 282: Please check any excess spaces.

In the text, instead of “Fig.”, “Figure” would be better.

Author Response

The revised manuscript contains important information and is better organized than in the first version. There are some linguistic improvements in the revised version.

In my opinion, the work is ready for printing after taking into account minor imperfections that I notice:

  • Line 202: In the figure caption, please note that there is no capital letter and unnecessary space.

Response:Thank you for your careful work. We have modified this figure caption.

  • Figure 7: Poor drawing quality. Although the axis descriptions should be improved.

Response:Thank you for your suggestion. We have improved the figure quality.

  • Line 282: Please check any excess spaces.

Response:Thank you for your careful work. We have reconfirmed the spaces in this sentence.

  • In the text, instead of “Fig.”, “Figure” would be better.

Response:Thank you for your careful work. We have modified these expressions in our manuscript.

Reviewer 3 Report (New Reviewer)

The paper studied the Global sea level changes using satellite altimetry and Grace satellite data to investigate the ocean thermal expansion. The authors concluded that sea level rise caused by the increase of water volume accounts for 62.7% of the total sea level rise. 

The research topic is interesting and of importance to readers. The introduction reads well and the authors have done a good  job in providing all the required information. However, I couldn’t see the innovation of the paper as compared to other studies  using altimetry and Grace satellite data. There are a number of research studies done using Grace satellite to assess the contribution of thermal expansion. The authors need to highlight their contributions better and state how their approach is different to others.

In my opinion, the data section could be shorten and rather explaining the basic explanation of each altimetry correction such as troposphere, ionosphere, atmosphere, sea state… authors need to present the specific model used for this purpose. 

The discussion could be improved and provide detailed comparison to similar studies using Grace data. 

There were also a few minor comments:

Please provide higher quality imagery/ figures.

Please make sure all the equations are cited properly. No references were provided for Equations 4, 5, 8, etc.

Please the consistent formatting, for example both JASON and Jason are used in the paper. 

Author Response

The paper studied the Global sea level changes using satellite altimetry and Grace satellite data to investigate the ocean thermal expansion. The authors concluded that sea level rise caused by the increase of water volume accounts for 62.7% of the total sea level rise. 

  • The research topic is interesting and of importance to readers. The introduction reads well and the authors have done a good job in providing all the required information. However, I couldn’t see the innovation of the paper as compared to other studies using altimetry and Grace satellite data. There are a number of research studies done using Grace satellite to assess the contribution of thermal expansion. The authors need to highlight their contributions better and state how their approach is different to others.

Response:GRACE satellite data are used to calculate the changes of global sea level caused by water increase in this study. Differed from the conventional method that uses the map data, this study adopts along track data to calculate the total sea level change with the help of satellite altimetry. Due to the use of interpolation, the map data itself may not be accurate. The sea level height obtained by using the along track data is closer to the reality. At the same time, this study unifies the calculation basis and processing methods of each error correction among altimetry satellite missions. In this way, for the same type of measurement error, such as sea state bias, the error caused by the use of different models will not be introduced.

  • In my opinion, the data section could be shorten and rather explaining the basic explanation of each altimetry correction such as troposphere, ionosphere, atmosphere, sea state… authors need to present the specific model used for this purpose.

Response:Thank you for your suggestion. Reference corrections were added in the text.

  • The discussion could be improved and provide detailed comparison to similar studies using Grace data.

Response:Thank you for your suggestion. In the discussion, a paragraph about the benefits of adopting the sea level calculation method proposed in this paper is added.

 There were also a few minor comments:

  • Please provide higher quality imagery/ figures.

Response:Thank you for your suggestion. The quality of all figures is updated in text.

  • Please make sure all the equations are cited properly. No references were provided for Equations 4, 5, 8, etc.

Response:Thank you for your suggestion. We have added several references.

  • Please the consistent formatting, for example both JASON and Jason are used in the paper.

Response: Thank you for your suggestion. Capitalizes the first letter.

Reviewer 4 Report (New Reviewer)

This paper examined the global sea level change using multiple data, which is interesting but needs some more explanations about the new findings of this paper.

Figure 3. Please add an explanation about the fusion and its advantage.

Figure 4. The global trend has been introduced many times in other literature. Can you also show a local trend that can be unique according to oceans?

L406. Global sea level change is very well known. The authors should focus on the original new finding in this paper. Can you explain this in more detail?

Author Response

This paper examined the global sea level change using multiple data, which is interesting but needs some more explanations about the new findings of this paper.

  • Figure 3. Please add an explanation about the fusion and its advantage.

Response: Our deepest gratitude goes to you for your careful work. We have add an explanation about the fusion and its advantage in the introduction.

  • Figure 4. The global trend has been introduced many times in other literature. Can you also show a local trend that can be unique according to oceans?

Response: Thank you for your suggestion. Lots of factors can affect regional sea-level variability, including winds and local currents that push water consistently toward or away from a particular shore. Due to the influence of the active structure, the coastal zone is increased or decreased, and the sea level underwent a larger variation. We are also working on correlation analysis of sea Level in offshore of China. But there was no immediate word on the outcome.

  • Global sea level change is very well known. The authors should focus on the original new finding in this paper. Can you explain this in more detail?

Response: Thank you for your suggestion. The original research methods and results of this study are given in detail in introduction and discussion. 

Reviewer 5 Report (New Reviewer)

I recommend this manuscript for publication however, the Figures should be improved, as they are not of good quality.

Author Response

  • I recommend this manuscript for publication however, the Figures should be improved, as they are not of good quality.

Response: Thank you for your suggestion. The quality of all figures is updated in text.

Round 2

Reviewer 4 Report (New Reviewer)

English language and style of the paper should be proofread by a native English speaker. Many mistakes are found, including Line 416 (analyzed the contribute of each) and Line 419 (sea level change are changing).

Author Response

English language and style of the paper should be proofread by a native English speaker. Many mistakes are found, including Line 416 (analyzed the contribute of each) and Line 419 (sea level change are changing).

Response: Thank you for your careful review. We apologize for the language problems in the manuscript. The language presentation was improved with assistance from a native English speaker with appropriate research background, so we hope it can meet the journal’s standard.

This manuscript is a resubmission of an earlier submission. The following is a list of the peer review reports and author responses from that submission.

Round 1

Reviewer 1 Report

Review for “Analysis of Global Sea Level Change Based on Multi-source Data”

  • There are similar research proposed with similar results, I did not see any significant difference in this paper comparison with other studies. NOAA published these results on line and can be used to distinguish the research results in this paper and compare with theirs. There is no significant improvement, in results, methods, or dataset used. The only new part is using the data from HY-2B, but I did not see the authors checking specifically the contribution of this satellite, which might make this paper unique to similar studies using Jason series and Gravity satellites. If the authors emphasis on the extension of the sea level rise time series with gravity measurements and steric height rise estimation, they should compare with this study: https://www.star.nesdis.noaa.gov/socd/lsa/SeaLevelRise/documents/NOAA_NESDIS_Sea_Level_Rise_Budget_Report_2014.pdf and similar studies. Did they conclude differently?  In the instruction section, the author did not bring up what the uniqueness and novelty of their study are.
  • Large amount of references in this paper are not peer-reviewed and in Chinese literature. While English abstract are available, the non-peer reviewed citation can discredit the quality of the paper.
  • It lacks of discussion for the contribution from mass monitoring, steric height changes and altimetry measurements. The land subsidy seems not in the interest in this paper, while it can be an important contribution for the sea level changes.
  • The writing of the paper needs improvements, some sentences are hard to understand and needs to be short and concise.
  • I suggest the authors revise this paper thoroughly and resubmit. 

Comments (not all, authors need to revise thoroughly to make the paper more readable):

  • Abstract: Line 11. Global sea level change and sea level rise are two different concepts, do not confuse sea level rise (can be associated with global warming) with sea level changes (seasonal variation, decadal variation etc).
  • Line 15, GRACE satellite data and in situ (or Argo float) temperature and salt data. Separate to two sentences, one for GRACE data and one for in situ data,  since you are talking about two things.
  • Line 25, these statement needs references, Church et al., 2011 can be good reference, also take a look at this website : https://www.climate.gov/news-features/understanding-climate/climate-change-global-sea-level
  • Line 33, not sure what you mean here ‘rest and property safety’
  • Line 34, reasons should be causes.
  • Line 35, The change of salinity of sea water can be caused by the fresh water input/output. It should be cautious on the role of the salinity when talking about the mean sea level rise. The main contributions of the global sea level changes: steric height due to thermal expansion (warming), the net fresh water input due to ice cap melting, and land subsidy. Check this paper:

Leuliette, E.W. The Balancing of the Sea-Level Budget. Curr Clim Change Rep 1, 185–191 (2015). https://doi.org/10.1007/s40641-015-0012-8

  • Line 36, in the ocean, what process can be attributed to infiltration?
  • Line 41, I did not see how sea water quality change affect the sea level, also the specific volume sea level change is not an appropriate word, why not use ‘steric sea level change’? as in the reference 2[Cazenave et al., 2003].  Too many references in non-English language is not appropriate here since some of them are not peer-reviewed and it is hard to find some of the references to check the contents.
  • Line 56, I believe it is Argo float instead of buoy. The buoy does not move much and Argo float moves along certain path in the ocean.
  • Line 60-75, it is a surprise there are no other referred literatures studying the global sea level rise from NOAA, AVISO, NASA and EUMETSAT, where the Altimetry and GRAVITY missions are for sea level change monitoring.
  • Line 76-78, the authors need to bring up what’s the motivation, and novelty of their study. Including the HY-2B can be a major contribution, if the authors can prove HY-2B measurements give rises to the similar trends/seasonal cycles to the global sea level changes from other satellites. Or if not , then state the problems in the measurement, calibration and processing.
  • Line 85, spell out GDR. Is this the final geophysical product? Why do the authors use only Jason series and HY-2B measurements? What are the difficulty in using the data from ENVISAT, SARAL/Altika?
  • How to understand the word ‘inversely’ here? Is the mass changes inversely related to the surface height changes?
  • Line 104-106, rephrase, not sure what’s the meaning here. Did you apply a coastal resolution improvement filter to the RL06M datasets?
  • 113, change ‘the Argo floats have been deployed in the global ocean,….providing a rich source of temperature and salinity measurement for the study…’.
  • Line 121, Table 2, are these standard World Ocean Atlas levels? Change ‘Number of Storeys’ to ‘level number’.
  • Line 124-125, Rephrase this sentence.’ Global sea level changes can be attribute to the sea water mass changes and steric sea level change’.
  • Line 127, when talking about salinity’s role in the steric height; do not confuse with the salinity changes from fresh water input.
  • Line 132, eq. 4, in both references, I did not see they mention the sea level changes from seawater quality.
  • Equation 2),3),4), have you done any of the mentioned corrections with GDR datasets in 4)? I believe the GDR datasets have already been corrected with the mentioned terms, atmospheric and tidal corrections. Please make it clear.
  • Line 158-162, break this long sentence into short, concise ones.
  • Line 166-169, not understandable. Since you use the regular lat/lon grids, the grid spacing (or area) changes with latitude. So a weighted average to derive the global mean is necessary.
  • Line 202, there is no need to discuss the dynamic model since the authors did not use them. How does the linear regression remove the trend uncertainties from the seasonal variation and long term climate signals?
  • Line 213, what’s Rong’s observation? The seasonal pattern should be consistent from all of the processing. What caused the seasonal variation in a globally averaged quatity?
  • Line 220, the acceleration is sensitive to the long term climate signals (decadal), is this increased acceleration due to decal variation (such as PDO) or simply algorithm caused difference? After all, the authors did not remove 2-7 years and longer term variations.  Change ‘the rate of rise of sea level change ‘ to ‘the rate of sea level rise’
  • Line 249, what is C20? Line 258, CRI?
  • Line 261, RL06 or RL06M?
  • Line 263, how many months of data are missing?
  • Line 292, looks the figure 6 can be removed.
  • Why is the sea level change from GRAVITY measurement called ‘steric sea level change’ in Figure 7? It looks the authors have a concept misunderstanding what is steric sea level changes. The GRAVITE series satellites monitor the gravity fields changes, mainly the mass changes, which is mostly related to net fresh water input into the ocean.
  • Line 306, reference to 1 km depth to derive the steric height might affect your results, generally 1.5km to 2000m are considered as the reference level with zero geostrophic current level (see Eric Leuliette , 2014). More discussion on this reference level is needed rather than a citation to a non-peer reviewed master degree thesis.
  • Line 313, P is not the pressure on the seafloor. It is calculated from density profiles to certain depth.
  • Line 315, it should be ‘one-atmosphere International Equation of State’. The one-atmosphere means one atmosphere standard pressure (meaning on the sea surface). Add the reference too.
  • Line 320, How did you do the interpolation? Sounds like extrapolation by using ‘to 2020’.
  • 322, linear regression and linear least squares are the same. 323, a linear model is fitted…
  • Line 330, when using least square, the interpolation is not needed if you specify the seasonal trigonometric cos function with linear model. There is no significant bias in Argo salinity data, I believe.
  • Line 340-348, break into short sentences. It is hard to capture what’s your point here.
  • Figure 8 can be removed and all the figures quality needs to be updated (increase font/contrast).
  • Line 350, change ‘measurements’ to ‘time series’.
  • Line 358-359, do not understand what ‘jointly responsible’ means here.
  • Table 4 and Table 4.1, listing of these numbers are not useful without describing what’s the difference in algorithm/processing.
  • Section 4.2 can be much simplified by just giving the correlation coefficients and its significance between two time series.
  • I am more interested in the trend uncertainties in the time series analysis rather than the correlation coefficients. The seasonal variability can explain most of the correlations, there is no surprise the correlation between any of the two time series in this paper is high. But since the authors want to derive the trend and acceleration, the uncertainties should be given. Are the trends reliable?
  • Line 403, change ‘combined’ to ‘used’. You did not combine all the data sets, just used different datasets to construct different time series for analysis.
  • Line 407, ‘impact’ to ‘contribute’.
  • I suggest the authors read through the NOAA website and other relevant literatures to enrich the discussion for different factors contributing to the global sea level rising.

Author Response

There are similar research proposed with similar results, I did not see any significant difference in this paper comparison with other studies. NOAA published these results on line and can be used to distinguish the research results in this paper and compare with theirs. There is no significant improvement, in results, methods, or dataset used. The only new part is using the data from HY-2B, but I did not see the authors checking specifically the contribution of this satellite, which might make this paper unique to similar studies using Jason series and Gravity satellites. If the authors emphasis on the extension of the sea level rise time series with gravity measurements and steric height rise estimation, they should compare with this study: https://www.star.nesdis.noaa.gov/socd/lsa/SeaLevelRise/documents/NOAA_NESDIS_Sea_Level_Rise_Budget_Report_2014.pdf and similar studies. Did they conclude differently?  In the instruction section, the author did not bring up what the uniqueness and novelty of their study are.

Response:Thank you for your careful read and thoughtful comments. Those comments are valuable and very helpful. We have read through comments carefully and have made corrections. As you note, NOAA have published the results of global sea level rise. In fact, in addition to NOAA, several institutions have also published similar results, such as NASA, CSIRO, Copernicus Marine Service, etc. These results are not exactly the same. The main reason is that the method and the data used are not exactly the same. For example, the global sea level rise results provided by AVISO to CMEMS are calculated based on map data, and the NOAA results are obtained based on tide gauge, satellite and other data. The production process of map data will destroy the measured value of satellite altimetry itself. The survey results of tide gauge data need to be corrected due to land subsidence. The calculation of global sea level rise introduced in the manuscript avoids the interference of untrue data as much as possible, and tries to calculate it entirely with satellite altimetry data.

  • Large amount of references in this paper are not peer-reviewed and in Chinese literature. While English abstract are available, the non-peer reviewed citation can discredit the quality of the paper.

Response:Thank you for your suggestion. We have added several references in English.

  • It lacks of discussion for the contribution from mass monitoring, steric height changes and altimetry measurements. The land subsidy seems not in the interest in this paper, while it can be an important contribution for the sea level changes.

Response:Section 4 of the manuscript gives the contribution of seawater mass and steric sea level change to sea level rise. Due to the lack of data, land subsidy was not discussed in this study.

  • The writing of the paper needs improvements, some sentences are hard to understand and needs to be short and concise.

Response:We apologize for the language problems in the original manuscript. The language presentation was improved with assistance from a native English speaker with appropriate research background. We agree with the comment and re-wrote the sentence in the revised manuscript.

  • I suggest the authors revise this paper thoroughly and resubmit. 

Comments (not all, authors need to revise thoroughly to make the paper more readable):

Response:We feel great thanks for your professional review work on our article. As you are concerned, there are several problems that need to be addressed. According to your nice suggestions, we have made extensive corrections to our previous draft; the detailed corrections are listed below.

  • Abstract: Line 11. Global sea level change and sea level rise are two different concepts, do not confuse sea level rise (can be associated with global warming) with sea level changes (seasonal variation, decadal variation etc).

Response:We are grateful for the suggestion. To be more clearly and in accordance with the reviewer concerns, we have replaced “global sea level change” with “global sea level rise”.

  • Line 15, GRACE satellite data and in situ (or Argo float) temperature and salt data. Separate to two sentences, one for GRACE data and one for in situ data,  since you are talking about two things.

Response:We deeply appreciate your suggestion. According to your comment, it has been divided it into two sentences.

  • Line 25, these statement needs references, Church et al., 2011 can be good reference, also take a look at this website : https://www.climate.gov/news-features/understanding-climate/climate-change-global-sea-level

Response: Thank you for your suggestion. Several references were added to the revised manuscript.

  • Line 33, not sure what you mean here ‘rest and property safety’

Response: I'm very sorry, this is a clerical error.

  • Line 34, reasons should be causes.

Response: Thank you for your careful work. We have modified this expression.

  • Line 35, The change of salinity of sea water can be caused by the fresh water input/output. It should be cautious on the role of the salinity when talking about the mean sea level rise. The main contributions of the global sea level changes: steric height due to thermal expansion (warming), the net fresh water input due to ice cap melting, and land subsidy. Check this paper:

Leuliette, E.W. The Balancing of the Sea-Level Budget. Curr Clim Change Rep 1, 185–191 (2015). https://doi.org/10.1007/s40641-015-0012-8

Response: Thank you for your kind advice.

  • Line 36, in the ocean, what process can be attributed to infiltration?

Response: Thank you for your hard work. The infiltration process in the ocean is not clear, so it is deleted in our manuscript.

  • Line 41, I did not see how sea water quality change affect the sea level, also the specific volume sea level change is not an appropriate word, why not use ‘steric sea level change’? as in the reference 2[Cazenave et al., 2003].  Too many references in non-English language is not appropriate here since some of them are not peer-reviewed and it is hard to find some of the references to check the contents.

Response: Thank you for your hard work. We revised the sentence with more accurate words. Peer reviewed English articles have been added to the references.

  • Line 56, I believe it is Argo float instead of buoy. The buoy does not move much and Argo float moves along certain path in the ocean.

Response: Thank you for your suggestion. Similar errors in the manuscript have been corrected.

  • Line 60-75, it is a surprise there are no other referred literatures studying the global sea level rise from NOAA, AVISO, NASA and EUMETSAT, where the Altimetry and GRAVITY missions are for sea level change monitoring.

Response: Thank you for the suggestions you kindly offered us. NOAA, CMEMS(AVISO), NASA, EUMETSAT and CRISO have indeed released products for global sea level rise. These organizations or institutions have made many contributions to the research of global sea level rise. Their works were added to the manuscript.

  • Line 76-78, the authors need to bring up what’s the motivation, and novelty of their study. Including the HY-2B can be a major contribution, if the authors can prove HY-2B measurements give rises to the similar trends/seasonal cycles to the global sea level changes from other satellites. Or if not , then state the problems in the measurement, calibration and processing.

Response: Thank you for your suggestions. To be more clearly and in accordance with the reviewer concerns, we have added a more detailed interpretation regarding our motivation in the text.

  • Line 85, spell out GDR. Is this the final geophysical product? Why do the authors use only Jason series and HY-2B measurements? What are the difficulty in using the data from ENVISAT, SARAL/Altika?

Response: Thank you for your suggestion and questions. GDR explained in the revised manuscript. Yes, GDR product is the final geophysical product. Only Jason series satellites and HY-2B satellites are selected in this study to prove that using the along track data, rather than grid data, is more suitable for the study of global sea level rise. The reason is that the along track data has not been interpolated in the meshing process, and the measurement accuracy has not been damaged.

  • How to understand the word ‘inversely’ here? Is the mass changes inversely related to the surface height changes?

Response: Thank you for your questions. The original manuscript is unclear here. We rephrased it.

  • Line 104-106, rephrase, not sure what’s the meaning here. Did you apply a coastal resolution improvement filter to the RL06M datasets?

Response: Thank you for your suggestion. The description of RL06M datasets has been rephrased.

  • 113, change ‘the Argo floats have been deployed in the global ocean,….providing a rich source of temperature and salinity measurement for the study…’.

Response: Thank you for your suggestion. The description of Argo floats has been rephrased.

  • Line 121, Table 2, are these standard World Ocean Atlas levels? Change ‘Number of Storeys’ to ‘level number’.

Response: The data we used are downloaded from the Copernicus marine environment monitoring website. The depth range of these data does not adopt standard world ocean atlas levels. Therefore, we listed depth ranges and standard depth levels numbers in detail in the manuscript.

  • Line 124-125, Rephrase this sentence.’ Global sea level changes can be attribute to the sea water mass changes and steric sea level change’.

Response: We deeply appreciate your suggestion. According to your comment, we have rephrased it.

  • Line 127, when talking about salinity’s role in the steric height; do not confuse with the salinity changes from fresh water input.

Response: Thank you for your remind.

  • Line 132, eq. 4, in both references, I did not see they mention the sea level changes from seawater quality.

Response: We have modified this expression throughout the text according to your comment.

  • Equation 2),3),4), have you done any of the mentioned corrections with GDR datasets in 4)? I believe the GDR datasets have already been corrected with the mentioned terms, atmospheric and tidal corrections. Please make it clear.

Response: Thank you for your good question. The Geophysical Data Record (GDR) product containing radar range, orbital altitude, and water vapor from the MWR as well as all geophysical corrections. However, SSH and SLA are not available in the product. Although we do not need to calculate any corrections, we need to calculate SSH and SLA using the dry troposphere correction, the wet troposphere correction, the sea state deviation correction, the tidal corrections et al.

  • Line 158-162, break this long sentence into short, concise ones.

Response: Thank you for your suggestion. According to your comment, we have rephrased it.

  • Line 166-169, not understandable. Since you use the regular lat/lon grids, the grid spacing (or area) changes with latitude. So a weighted average to derive the global mean is necessary.

Response: Thank you for your suggestion. According to your comment, we have rephrased it.

  • Line 202, there is no need to discuss the dynamic model since the authors did not use them. How does the linear regression remove the trend uncertainties from the seasonal variation and long term climate signals?

Response: Thank you for your comment. The seasonal variation and long term climate signals are close to periodicity. When linear regression is used, the periodic signal will not affect the acquisition of sea level rising trend.

  • Line 213, what’s Rong’s observation? The seasonal pattern should be consistent from all of the processing. What caused the seasonal variation in a globally averaged quatity?

Response: Only the English abstract is available for Rong’s paper. Readers who don't understand Chinese can't understand the details. This comparison result is removed.

  • Line 220, the acceleration is sensitive to the long term climate signals (decadal), is this increased acceleration due to decal variation (such as PDO) or simply algorithm caused difference? After all, the authors did not remove 2-7 years and longer term variations.  Change ‘the rate of rise of sea level change ‘ to ‘the rate of sea level rise’

Response: Thank you for your questions. PDO, AMO and other Decal variations will cause periodic sea level changes. From 2002 to 2020, PDO is in the cold phase which cause sea level to fall. This paper only gives the rise of sea level, and does not analyze the impact of decadal variation such as PDO. ‘the rate of rise of sea level change ‘ has been changed to ‘the rate of sea level rise’.

  • Line 249, what is C20? Line 258, CRI?

Response: Thank you for your questions. Two abbreviations are explained in the text.

  • Line 261, RL06 or RL06M?

Response: Thank you for your careful review. The revised manuscript has been corrected.

  • Line 263, how many months of data are missing?

Response: About 26 months.

The month marked in red is the month with missing data.

  • Line 292, looks the figure 6 can be removed.

Response: This picture is really unnecessary.

  • Why is the sea level change from GRAVITY measurement called ‘steric sea level change’ in Figure 7? It looks the authors have a concept misunderstanding what is steric sea level changes. The GRAVITE series satellites monitor the gravity fields changes, mainly the mass changes, which is mostly related to net fresh water input into the ocean.

Response: Thank you for underlining this deficiency. Corrections have been made in the manuscript.

  • Line 306, reference to 1 km depth to derive the steric height might affect your results, generally 1.5km to 2000m are considered as the reference level with zero geostrophic current level (see Eric Leuliette , 2014). More discussion on this reference level is needed rather than a citation to a non-peer reviewed master degree thesis.

Response: Thank you for your suggestions. More detailed discussion on the reference level was added.

  • Line 313, P is not the pressure on the seafloor. It is calculated from density profiles to certain depth.

Response: Thank you for underlining this deficiency. It is applied pressure.

  • Line 315, it should be ‘one-atmosphere International Equation of State’. The one-atmosphere means one atmosphere standard pressure (meaning on the sea surface). Add the reference too.

Response: Thank you for suggestion. It has been revised.

  • Line 320, How did you do the interpolation? Sounds like extrapolation by using ‘to 2020’.

Response: Thank you for suggestion. It is really extrapolated to 2020.

  • 322, linear regression and linear least squares are the same. 323, a linear model is fitted…

Response: Thank you for suggestion.

  • Line 330, when using least square, the interpolation is not needed if you specify the seasonal trigonometric cos function with linear model. There is no significant bias in Argo salinity data, I believe.

Response: Thank you for your precious comments and advice. Since 2016, salinity data measured by some Argo floats have deviation. Interpolation is used to deal with this deviation.

  • Line 340-348, break into short sentences. It is hard to capture what’s your point here.

Response: Thank you for suggestion. As suggested, we have broken the long sentence into several short sentences.

  • Figure 8 can be removed and all the figures quality needs to be updated (increase font/contrast).

Response: Thank you for suggestion. We have removed it. And the quality of all figures is updated in text.

  • Line 350, change ‘measurements’ to ‘time series’.

Response: Thank you for suggestion.

  • Line 358-359, do not understand what ‘jointly responsible’ means here.

Response: To be more clearly, we rewrote this sentence.

  • Table 4 and Table 4.1, listing of these numbers are not useful without describing what’s the difference in algorithm/processing.

Response: Table 4 has been removed in the revised manuscript.

  • Section 4.2 can be much simplified by just giving the correlation coefficients and its significance between two time series. I am more interested in the trend uncertainties in the time series analysis rather than the correlation coefficients. The seasonal variability can explain most of the correlations, there is no surprise the correlation between any of the two time series in this paper is high. But since the authors want to derive the trend and acceleration, the uncertainties should be given. Are the trends reliable?

Response: Thank you for your precious comments and advice. In Section 4.2, we mainly want to give two results. One is that there is a good correlation between the sea-level rise obtained using the along track data and the grid data, which proves that the method proposed in this paper is feasible and the sea-level rise trend obtained is reliable. The other is that the sea level rise obtained from satellite altimetry data has a good correlation with the sum of sea water master change and steric sea level change.

  • Line 403, change ‘combined’ to ‘used’. You did not combine all the data sets, just used different datasets to construct different time series for analysis.

Response: Thank you for suggestion. It has been revised.

  • Line 407, ‘impact’ to ‘contribute’.

Response: Thank you for suggestion. It has been revised.

  • I suggest the authors read through the NOAA website and other relevant literatures to enrich the discussion for different factors contributing to the global sea level rising.

Response: Thank you for suggestion.

Reviewer 2 Report

The results of the analyzes of multi-annual sea level data are interesting. The authors are looking for the possibility of linking the increase in the volume of the ocean with the increase of its temperature and the increase in runoff from land and ice caps on both poles of the Earth. The authors share with the manuscript reader the awareness that the elaboration of the current measurement data does not yet allow for the formulation of categorical statements as to the cause of the sea level rise.
Some pictures (e.g. 8 and 9) require correction of the inscriptions (they are blurry).
Are the authors sure the final chapter is to be called "Conclusions and Discussion" and not "Discussion and Conclusions"?

Author Response

The results of the analyzes of multi-annual sea level data are interesting. The authors are looking for the possibility of linking the increase in the volume of the ocean with the increase of its temperature and the increase in runoff from land and ice caps on both poles of the Earth. The authors share with the manuscript reader the awareness that the elaboration of the current measurement data does not yet allow for the formulation of categorical statements as to the cause of the sea level rise.

Response: We appreciate the reviewer’s positive evaluation of our work.
Some pictures (e.g. 8 and 9) require correction of the inscriptions (they are blurry).

Response: Thank you for your suggestion. The blur picture has been updated.
Are the authors sure the final chapter is to be called "Conclusions and Discussion" and not "Discussion and Conclusions"?

Response: Our deepest gratitude goes to you for your careful work. We have modified this expression.

Reviewer 3 Report

There are fundamental uncertainties in the rates, amplitudes, and mechanisms for the global sea level change. Based on the altimeter, the characteristics of the inter-annual variability of the global sea level are analyzed in this study. We all know that thermal expansion of seawater and the melting of glaciers and ice sheets on land due to global warming contribute to the accelerated rise in global sea level. However, it is difficult to quantitatively analyze the rate of sea level change using satellite and other observation data. GRACE satellite data and temperature salt data are used to analyze the changes of global sea level caused by water increase and ocean thermal expansion. The methods and conclusions are novel and the paper is well structured and written. So, my suggestion is minor revision. 

Some suggestions for the author:

  1. There may be plenty of minor faults such as font size in Table 2.
  2. In addition, there are obvious differences between altimeter measurement and grace plus Argo measurement after 2015. In addition to the reasons mentioned in the paper, is it possible that new factors leading to sea level rise have emerged in recent years?

Author Response

There are fundamental uncertainties in the rates, amplitudes, and mechanisms for the global sea level change. Based on the altimeter, the characteristics of the inter-annual variability of the global sea level are analyzed in this study. We all know that thermal expansion of seawater and the melting of glaciers and ice sheets on land due to global warming contribute to the accelerated rise in global sea level. However, it is difficult to quantitatively analyze the rate of sea level change using satellite and other observation data. GRACE satellite data and temperature salt data are used to analyze the changes of global sea level caused by water increase and ocean thermal expansion. The methods and conclusions are novel and the paper is well structured and written. So, my suggestion is minor revision.

Response: We appreciate your positive evaluation of our work.

Some suggestions for the author:

There may be plenty of minor faults such as font size in Table 2.

Response: Thank you for your careful work. We have revised the manuscript carefully.

In addition, there are obvious differences between altimeter measurement and grace plus Argo measurement after 2015. In addition to the reasons mentioned in the paper, is it possible that new factors leading to sea level rise have emerged in recent years?

Response: Thank you for your question. There are many factors affecting sea level rise, such as underground water, dam, glaciers and ice sheets, et al. Some know, some don't. But global warming is a major factor in sea-level rise.

Round 2

Reviewer 1 Report

This paper has been greatly revised. I think it can be published. I have only a few minor comments:

1.)    Line 72, list references.

2.)    Line 114, change 'salt' to 'salinity'. (also check other places in text)

3.)    Line 393, what do you mean ‘mass and spatial resolution’?
